# Fishermen Interviews: A Cost-Effective Tool for Evaluating the Impact of Fisheries on Vulnerable Sea Turtles in Tunisia and Identifying Levers of Mitigation

**DOI:** 10.3390/ani13091535

**Published:** 2023-05-04

**Authors:** Maissa Louhichi, Alexandre Girard, Imed Jribi

**Affiliations:** 1BIOME Lab, Sfax Faculty of Sciences, Univerity of Sfax P.O. Box 1171, Sfax 3000, Tunisia; 2PatriNat, Muséum National d’Histoire Naturelle, rue Buffon, 75005 Paris, France

**Keywords:** sea turtles, sustainable fisheries, bycatch, Tunisia, vulnerable species, Mediterranean Sea, Africa

## Abstract

**Simple Summary:**

Fisheries represent one of the main threats to sea turtles in the Mediterranean Sea, but the level of interaction of sea turtles with fisheries remains poorly assessed. Using interviews with fishermen as its basis, this article assesses the interaction of sea turtles with fisheries in Tunisia. The interview results allowed the calculation of the bycatch per unit effort (BPUE) for the most common fishing gear used in Tunisia, and they showed that ray nets, trawls, and pelagic longlines represent the greatest threat. This confirms the value of the interview approach when assessing the impact of fishing gear on sea turtles. In addition, the analysis of the bycatch risk factors based on the interview data provides a critical early step towards a reduction in the fisheries’ impact on sea turtles in Tunisia.

**Abstract:**

Fisheries bycatch is considered one of the main threats to sea turtles. To provide an updated assessment of the bycatch impact on loggerhead turtles in Tunisia, 483 interviews were conducted with fishermen in 19 ports along the Gulfs of Gabes and Hammamet. The interview questions included data on vessel and fishing gear characteristics, monthly fishing effort, and monthly sea turtle bycatch of the last year. Results revealed that sea turtle bycatch per unit effort (BPUE) was the highest for ray nets (0.73 turtles/vessel/day, 95% CI [0.02–1.15]), followed by pelagic longlines (0.6 turtles/vessel/day, 95% CI [0–4.23]) and shark nets (0.4 turtles/vessel/day, 95% CI [0–2.12]). However, due to the trawlers’ high fishing effort, the cumulative impact of the trawl nets was the highest with an estimated number of 11,740 (95% CI [0–41,525.75]) turtles caught per year in Tunisia. Factors influencing the bycatch risk were gear characteristics (mesh size for nets, hook size for longlines, vertical opening for trawls), as well as season and location of operations. These factors will guide the research of mitigation solutions. The interviews with fishermen proved to be a cost-effective approach for the rapid assessment of sea turtle interactions with fisheries in the context of Tunisia.

## 1. Introduction

In the Mediterranean Sea, the fisheries are commonly multi-species and multi-gear. Both artisanal and large-scale commercial fisheries contribute to the overexploitation of the Mediterranean Sea fish resources [1,2]. The fisheries are also considered to be one of the main threats to the vulnerable species of marine megafauna, such as sharks, cetaceans, and sea turtles, through the direct mortalities and injuries associated with incidental bycatch [3,4,5,6]. The Mediterranean Sea is host to three species of sea turtles (the loggerhead turtle (*Caretta caretta*), green turtle (*Chelonia mydas*), and leatherback turtle (*Dermochelys coriacea)*). The loggerhead turtle (*Caretta caretta*, [7]) is the most abundant [6,8,9] and uses both the oceanic and neritic zones [5,10]. Loggerhead turtles use various foraging and reproductive areas and migratory pathways throughout the Mediterranean Sea; this often results in co-occurrence on productive commercial fishing grounds where they are susceptible to bycatch [5,10,11,12,13,14]. The major loggerhead neritic foraging grounds are located along the southeast coast of Turkey, in the Gulf of Gabes [10,11]; in the north Adriatic Sea [12], off the Egyptian coast [5]; and off the Spanish coast [13]. The major nesting beaches for the loggerhead are in Greece, Turkey, Cyprus, and Libya, with low nesting activities recorded in Egypt, Syria, Lebanon, Tunisia, and Italy [14]. There is no specific fishery or fishing gear directly targeting sea turtles, but due to their habits (e.g., breeding and feeding migrations), loggerhead turtles interact with several types of fishing gear, including demersal and pelagic trawls, set nets, and longlines [15,16].

A high level of uncertainty remains regarding the number of turtles caught and their mortality rates. According to the available literature, the turtle bycatch in the Mediterranean Sea is likely to be high: approximately 132,000 per year, with a high number of deaths, probably over 44,000 yearly [5,8,9,17]. Despite the risk of bycatch, in 2015 the IUCN listed the Mediterranean loggerhead population as being at the level of “Least Concern” [18]. More data, including those related to affected life stages and common locations, are currently required to evaluate the threat of bycatch to the longevity of this population.

The most commonly used types of fishing gear in Tunisia are set nets, trawls, and longlines. Until now, bycatch assessments were restricted in Tunisia to the Gulf of Gabes. This gulf, which is considered to be a biodiversity hotspot and an important loggerhead foraging and wintering area, is also the most important fishery area of the Tunisian fishing fleet. Studies conducted in the Gulf of Gabes confirm that trawls [10], longlines [19,20,21], and gillnets [22,23,24,25] are a major threat to sea turtles. However, the assessments of the interaction level (sea turtle bycatch per unit effort, BPUE) and the fisheries’ impact in terms of death and loss of survival are limited and old. The assessments of the fisheries’ impact need to be improved, extended, and updated. It is also crucial to rank the fishery threat to sea turtles by measuring and comparing the impact of the most common fishing gear on sea turtles in Tunisia.

To provide an updated assessment of the bycatch threat to loggerhead turtles in Tunisia, this study conducted interviews with fishermen using a range of gear types in all major Tunisian ports.

In Tunisia, as in the rest of the Mediterranean basin, several mitigation studies have experimented with the use of circle hooks for longlines, light devices for set nets, and TEDs for trawlers. No requirements to use these have been put in place, and the results are still controversial.

In addition, we assessed the bycatch risk factors. Finally, based on the results, we propose levers of mitigation that can improve the sustainability of the fisheries in Tunisia.

## 2. Materials and Methods

### 2.1. Interviews—Questionnaires

Interviews were conducted systematically in 19 out of the 33 ports recognized by the Fishing Ports and Facilities Agency along the coast from Cap-Bon, on the northern coast of Tunisia, to the Libyan border in the south (Figure 1).

The interviews were generally conducted directly with the captains of the fishing vessels. In the case that the captain could not be reached, the interview was carried out with another member of the crew. In all cases, one single fisherman was interviewed per vessel. The fishermen were generally interviewed in ports onboard their vessels. In some cases, the interviews were performed at the harbor master’s office or during meetings of the fishermen’s association.

All necessary permits were obtained for the field studies described. The interviewees were informed of the purpose of the study and that the data collected would remain confidential.

The data collected through the questionnaires were complemented by additional information, such as that regarding fleet and gear inventories, obtained from the Fisheries Regional Administrations (i.e., the General Headquarters of Fisheries).

A single questionnaire was designed to be completed in 20 min; it consisted of 4 sections: Characteristics of vessels involved: length, tonnage, number of fishermen, and engine power.Fishing gear used, including information about the seasons and gear characteristics (for nets: type of nets [correlated to mesh size], depth, distance between floats, and distance between sinkers; for longlines: longline type (benthic vs. pelagic), number of floats/km, number of sinkers/km, number of buoys/km, hook size, diameter and length of the mainline, and bait types; for trawls: otter board length, arm length, size of the cul-de-sac meshes, size of the vertical opening, length of the head rope, and length of the foot rope).For every gear type, the fishing effort was expressed in terms of the number of days per month at sea and the number of fishing operations per fishing trip for the last twelve months.For every gear type, information was given about the monthly frequency of the sea turtle bycatch and the mortality rate over the last twelve months; additional information was given about the bycatch of other animals in the same period, such as elasmobranchs, cetaceans, and birds.

The vessels to be included in the study were identified via opportunistic sampling. The number of interviews for each port was weighted according to the fleet’s relative importance based on national statistics. In each port, particular attention was paid to ensuring that the interviews represented the proportions and the diversity of the fishing gear. The most common types of fishing gear used along the coast of Tunisia were represented in the survey. The vessels interviewed were using trawls, pelagic longlines, bottom longlines, and different types of nets. 

### 2.2. Data Analysis

To allow for a comparison of the types of fishing gear, the fishing effort was expressed as the number of fishing days at sea, as obtained from the fishermen’s interviews. The mean sea turtle bycatch per unit effort (BPUE), according to the fishing gear, was expressed as the number of turtles caught per vessel per day. 

When available, the national fishery statistics were related to the BPUE to give an estimate of the global annual impact of one gear métier on sea turtles in Tunisia. A rough estimate of the global impact of fishing gear was subsequently obtained by multiplying the yearly mean number of sea turtles captured per vessel by the number of vessels recorded in the Tunisian national statistics.

For each major type of gear (longlines, nets, and trawls), the generalized linear model (GLM) [26,27] and the generalized linear mixed model (GLMM) were used to establish the relationships between the sea turtle bycatch and the different covariates related to the gear characteristics, fishing operations (including fishing efforts), and seasonal and geographical variation. 

The variables were assessed for normality and were transformed (e.g., by log transformation), when appropriate, to satisfy the assumptions of these models.

The covariates selected to be included in the models were those that potentially affected the sea turtle bycatch, based on the literature and studies formerly conducted in Tunisia [10,19,20,21,22,23,24,25]. The fishing vessels and ports were included as random factors in the mixed models. 

The models were examined for overdispersion [28] and zero inflation. Various theoretical distributions, including the Poisson distribution, negative binomial, zero-inflated Poisson, and zero-inflated binomial, were tested to model the bycatch quantity.

The models were run using R software (version 3.6.3), with the GLM function [29] in the MASS package [30], along with the associated methods for diagnostics and inference. The GLM and GLMM, which allow zero-inflated distribution, were run using the hurdle and zero-inflation functions within the pscl package [31] and the glmmTMB function within the glmmTMB package.

The model selection was performed following a stepwise selection using Akaike’s information criterion (AIC) [32].

## 3. Results

In total, 483 interviews were conducted in 19 ports from May to November 2017 (Figure 1).

### 3.1. Relative Impact on Sea Turtles and BPUE According to the Fishing Gear 

The BPUE (Figure 2) was the highest for ray nets (0.73 turtles/vessel/day, 95% CI [0.02–1.15]), followed by pelagic longlines (0.6 turtles/vessel/day, 95% CI [0–4.23]) and shark nets (0.4 turtles/vessel/day, 95% CI [0–2.12]). Benthic trawls (0.17 turtles/vessel/day, 95% CI [0–0.55]) and benthic longlines (0.13, 95% CI [0–1.23])) exhibited a lower BPUE. 

The other nets (encircling nets, bony fish gillnets, and trammel nets) exhibited a BPUE of less than 0.1.

### 3.2. Effect of Cofactors on the Level of Sea Turtle Bycatch Risk for Each Gear Type

#### 3.2.1. Fishing Nets

The model that best described the observed number of sea turtles caught per month for the vessels equipped with fishing nets was the zero-inflated negative binomial GLMM, which included the port and vessel units as random factors. Table 1 is an extract displaying only the significant effects; the complete version of Table 1 is available in the Appendix A. For the model selection results based on AIC, see Appendix A.

In the counting part of the ZINB model, the type of fishing net affected the monthly number of sea turtles caught per vessel; trammel nets and ray nets (Garrassia) had significant effects. The fishing effort, expressed as log (Effort operation month Length-Number of nets), also had a significant effect. The pseudo-Fourier variable accounting for the seasonality of the captures, combined with the particular gulf, was highly significant. 

The predicted mean number of loggerhead turtles caught with fishing nets per month per vessel was higher in the Gulf of Gabes compared to the Gulf of Hammamet. Nevertheless, the 95% confidence intervals overlap; the number of turtles caught by fishing nets in relation to the gulf was, therefore, not significantly different. The mean number of loggerhead turtles caught per month in the Gulf of Gabes varied between 1.17 and 3.45, and in the Gulf of Hammamet, it varied between 0.53 and 2.25 (Figure 3).

The sea turtle bycatch by fishing nets occurred throughout the year with an increase in the spring and summer (April–August) (Figure 4). The seasonality of the bycatch was slightly different in the two gulfs: the predicted peak of the sea turtle bycatch occurred earlier (May) in the Gulf of Gabes compared to the Gulf of Hammamet (June) (Figure 4).

The distance between the lead weights on the footrope was useful for the model of the observed data, but its marginal effect on the predicted sea turtle bycatch quantity was not significant.

The effect of mesh sizes on the predicted sea turtle bycatch quantity was not accessible in the model since the size of the mesh was dependent on the type of fishing net. However, large-mesh fishing nets proved to be the ones with a higher predicted sea turtle bycatch quantity (Figure 5).

#### 3.2.2. Longlines

The model that best described the monthly number of sea turtles caught by vessels equipped with longlines was the zero-inflated GLMM, which included the port and vessel units as random factors. Table 2 is an extract displaying only the significant effects; the complete version of Table 2 is available in the Appendix A. For the model selection results based on AIC, see Appendix A.

The longline type, hook size, and seasonality, which are interwoven with the identity of the gulfs, exhibited significant effects on the predicted number of sea turtles caught. In the best selected model, the longline type had a significant effect on zero inflation. 

The predicted mean number of sea turtles caught was slightly higher in the Gulf of Gabes (mean = 1.01 95% CI [0.17, 5.94]) than in the Gulf of Hammamet (0.58 [0.14, 2.38]), but not significantly different (Figure 6). 

The number of loggerhead turtles caught varied depending on the longline type. The mean predicted number of sea turtles captured by longline vessels within a month was higher for pelagic longlines 6.70 (95% CI 1.26, 35.76) than benthic longlines 1.01 (95% CI 0.17, 5.94) (Figure 6).

The sea turtle bycatch by longlines occurred throughout the year with an increase in the spring and summer (April–August) (Figure 7). The seasonality of the bycatch was different in the two gulfs: the predicted peak of the sea turtle bycatch occurred earlier (spring) in the Gulf of Gabes compared to the Gulf of Hammamet (summer) for both types of longline.

In Tunisia, hook size is designated by numbering according to the SUN nomenclature (ISO 1837:2003). Hook size also affected the sea turtle bycatch quantity per month per vessel. The marginal effect curves exhibited a peak for medium-size hooks, number #5, for which the distance between the hook shank and the point is 1.8 cm (Figure 8).

#### 3.2.3. Trawls

Based on AIC, the model that best described the monthly number of sea turtles caught by trawlers was a hurdle-type zero-inflated GLMM (with the zero figures being adjusted independently and the negative binomial theoretical distribution of the GLMM being zero-truncated), which included the vessel ID unit as a random factor. Table 3 is an extract displaying only the significant effects; the complete version of Table 3 is available in the Appendix A. For the model selection results based on AIC, see Appendix A

When combined, the location (Gulf of Hammamet vs. Gulf of Gabes) and the seasons had a significant effect on the monthly bycatch quantity by trawlers. In addition to this, some trawl characteristics, including footrope and headrope lengths, also had an effect.

Other gear characteristics, such as trawl arm length, panel length, bottom mesh size, and haul duration, were useful with regard to the quality of the fit, but they had no significant effect on the response variable.

The monthly sea turtle captures per trawler also varied according to the season, with an increase in spring and summer (March–August) for both areas (Figure 9). The seasonality of the trawlers’ bycatch was slightly different in the two gulfs: the predicted peak of the sea turtle bycatch occurred earlier (May) in the Gulf of Gabes compared to that in the Gulf of Hammamet (June) (Figure 9).

The difference in the monthly predicted number of sea turtles caught was significant between the two gulfs. The predicted monthly mean number of sea turtle captures per trawler was 3.93 (95% CI [2.28–6.78]) for the Gulf of Gabes and 0.83 (95% CI [0.32–2.14]) for the Gulf of Hammamet.

The trawl characteristics also significantly influenced the predicted number of sea turtles caught by trawlers (Figure 10).

The monthly sea turtle bycatch increased with the length of the footrope (Figure 10a).

Conversely, the sea turtle bycatch decreased when the headrope length increased (Figure 10b).

## 4. Discussion

The interviews in the ports proved to be a cost-effective approach for the rapid assessment of the interaction between the fisheries and sea turtles in Tunisia. The highest BPUE was observed for shark and ray nets. However, due to the high level of fishing efforts by the trawlers, the cumulative impact of the trawl nets was the highest. A further GLMM analysis of the interview data revealed the major trends guiding the risk of bycatch and helped to identify the promising levers of mitigation. The interactions were higher in the Gulf of Gabes compared to the Gulf of Hammamet, particularly with regard to the trawlers, and the bycatch quantity was seasonal, with a peak during summer. The bycatch seasonality also differed significantly between the two gulfs. In addition, some gear characteristics, including the shape of the trawl’s vertical opening, the larger meshes in the gillnets, and the pelagic longlines with medium-size hooks, had a significant influence on the sea turtle bycatch rate. Those findings pave the way for mitigation solutions and evidence-based management, including seasonal closures, as well as modification of the gear characteristics.

Interest in the interview approach for assessing fisheries bycatch

The interview approach was recommended by the FAO for fishery assessments [33]. This study relies on the standards proposed by the FAO, and it demonstrates the efficiency and rapidity of this approach for the first-line assessment of fisheries and their interactions with sea turtles. Collecting the monthly sea turtle bycatch quantities and the associated fishing efforts produced a consistent BPUE estimate for the most frequently used fishing gear in operation in Tunisia. In addition, the data collection on the fishing gear characteristics, seasonality, and location of the fishing event allowed us to analyze the factors influencing the sea turtle bycatch rate.

Impact of Tunisian fisheries on sea turtles

The BPUE estimated from the interviews in Tunisia was high and consistent with the bycatch rates previously reported along the North African continental shelf of Tunisia [10,19,20,21,24,34], Egypt [35], the Levantine basin [36], the western coast of Sardinia [37] and the Balearic Islands [38]. This high catch rate can be related to the abundance of sea turtles, especially loggerhead turtles, in the Mediterranean Sea; the number of turtles is estimated to be from 1,197,087 individuals (95% CI: 805,658–1,732,675) to 2,364,843 individuals (95% CI: 1,611,085–3,376,104) [8].

The results of this study reveal that the BPUE was the highest for ray nets in Tunisia, followed by pelagic longlines, shark nets, benthic trawls, and benthic longlines (Figure 4 and Table 3). The other nets (encircling nets, bony fish gillnets, and trammel nets) exhibited a BPUE lower than 0.1. However, in terms of cumulative bycatch for one fishing vessel operating for one year, the trawls appeared to be one of the two gear types impacting the fisheries the most. While the trawl BPUE appears relatively low (0.17, 95% CI [0–0.55]) compared to the set nets, the high fishing effort employed by each trawler leads to trawlers capturing the highest number of sea turtles (mean: 30.18 turtle/vessel/year 95% CI [0–106.75]. The number of trawlers in both the Gulf of Gabes and the Gulf of Hammamet is 389, according to the Tunisian Fishery Ministry statistic [33]. A rough estimate of the national impact of trawling on sea turtles can be proposed: 30.18 * 389 = 11,740.02 (95% CI [0–41,525.75]) turtles are caught by trawlers per year in Tunisia. The results of bycatch studies in Italy similarly concluded that trawlers exhibit the highest impact [39].

The intense interaction between sea turtles and bottom trawlers in the Gulf of Gabes has been described previously [10,17]. According to these publications, the Tunisian sea turtle bycatch rate by trawlers is considered to be among the highest in the Mediterranean. The present study expands the coverage to the Gulf of Hammamet, and it reveals that the trawlers’ interaction with sea turtles is higher in the Gulf of Gabes than in the Gulf of Hammamet. The national impact of benthic trawling is high in both gulfs, with an estimated yearly mean number of 11,740 sea turtle captures in Tunisia. Casale [17] estimated that there were over 132,000 sea turtle captures in the Mediterranean annually by all the examples of fishing gear combined, 39,000 of which were caused by trawlers. More recent studies in specific areas suggest that these figures are underestimated [39], and the trawler bycatch estimate from our study also confirms that the figures are underestimated.

For longlines, the number of loggerhead turtles caught varies depending on the longline type. The mean number of captures for pelagic longlines per year per vessel is 6.70 turtles/year/vessel (95% CI [1.26–35.76]) and 1.01 turtles/year/vessel (95% CI [0.17–5.94]) for bottom longlines. However, the fishing effort is higher for bottom longlines than pelagic longlines, and the mortality risk is higher with bottom longlines. The catch rate by pelagic longlines observed in Tunisia is among the highest recorded in the Mediterranean Sea [26]. The use of pelagic longlines increases the catch rate risk since loggerhead turtles occur more frequently at depths of less than 50 m [40].

Mortality concurrent with bycatch

Beyond the bycatch rate, the impact of fisheries needs to be assessed in terms of the sea turtle mortality concurrent with bycatch, including both the direct mortality from bycatch and the possible survival reduction subsequent to release. Even though mortality and the factors influencing it were not the focus of this study, the mortality risk concurrent with bycatch can be ranked according to gear types and practices. Drowning due to forced apnea is the main reason for the sea turtle mortality caused by fishing gear; the animal, once caught in the gear, cannot reach the surface to breathe [40]. Gillnets are left at sea for many hours (sometimes many days); this is well beyond the tolerance range of turtles, causing a high mortality rate. According to fishermen and our own observations, the ray net, which is left for many hours and sometimes many days in the water, causes a high mortality rate. The impact of ray nets in terms of sea turtle mortality may, therefore, be among the highest. In the same way, the mortality rate is higher with benthic longlines compared to pelagic longlines since captured turtles can reach the surface to breathe in the latter [19]. However, even if a turtle survives and is freed, there still may be delayed mortality if the fisherman does not free the turtle from all the ropes of the net, which can cause serious injury and necrosis [40].

Turtle species and size classes interacting with fisheries

All the sea turtles observed were chelonids (scuted sea turtles), but species determination is difficult in the context of fishermen interviews. Two chelonid species occur in Tunisia: the loggerhead turtle (*Caretta caretta*) and the green turtle (*Chelonia mydas*). However, loggerhead turtles represent the large majority and green turtles are rare in Tunisian waters [10,19,24,41]. Therefore, the sea turtles observed as bycatch in the current study were alleged to be loggerhead turtles (*Caretta caretta*). Loggerhead turtles are more likely to feed on benthic prey when they are larger than when they are smaller. Therefore, trawlers fishing close to the sea bottom will capture turtles of a relatively large size. Bottom-trawling will, therefore, have a stronger impact on the sea turtle population since a larger (older) specimen makes a greater contribution to the demographic growth of the population to which it belongs [42,43,44,45].

Factors influencing the bycatch risk

Numerous factors, including the size of the fishing gear, operational depth, haul duration, speed, time of day, and season of fishing, differ among fisheries, among vessels from the same fishery, and even between trips or single operations of the same vessel, and they may have an impact on the various aspects of the turtle bycatch and mortality [17]. Other factors influencing the sea turtle catch rates and associated mortality are more closely tied to turtle biology (body size, life cycle phase, temperature, etc.). The GLMM results from this study provide an insight into these factors.

Influence of fishing location

Sea turtles are not uniformly distributed, and, for a given fishing effort with a given gear type, the catch rates will increase with the sea turtle concentration in the fishing location. The bycatch risk is high when a high fishing effort meets a high sea turtle concentration. The Gulf of Gabes illustrates such an overlap. It has been shown to be an important foraging and wintering area in the Mediterranean [40,46,47,48,49,50,51], and the concomitant high number of trawlers operating in the area explains the significantly higher impact of trawlers on the turtles in the Gulf of Gabes compared to the Gulf of Hammamet (Figure 9). The water column also plays an important role in the level of interaction; the literature reports that most of the captures by trawl nets occur in shallow waters, in areas where the water column is less than 50 m, or even 20 m, in depth [52,53,54,55]. The wide continental shelf extending east of Tunisia makes sea turtles more vulnerable in the two gulfs, particularly in the Gulf of Gabes.

Seasonal variations

Sea turtles are migratory species. Depending on environmental and biological factors, such as sea temperature, food availability, and reproduction cycles, sea turtles can travel great distances. Due to seasonal migrations between winter and summer and reproductive migrations [56], the abundance of turtles and, consequently, the bycatch rates might exhibit seasonal variation. The present study’s results show the seasonal variation of the bycatch, with a peak in June–July and the lowest level in November–December for the fisheries operating in the Gulf of Hammamet no matter what gear is used. For the vessels operating in the Gulf of Gabes, the bycatch peak occurs earlier: in March for longliners and in May for trawlers and set netters. The lowest level occurs during August–September for longliners and in November for trawlers and set netters.

The observed variations are consistent with the seasonal movement of loggerhead turtles in Tunisia, with the largest number of them concentrated on the continental shelves during the winter and then leaving in summer to nest in the eastern Mediterranean. The Gulf of Hammamet’s curves, which peak in June and July (Figure 9) might be due to the loggerhead movement for reproduction.

Gear characteristics influencing the sea turtle bycatch rate

The analysis of the factors influencing the sea turtle bycatch in trawls shows that some trawl characteristics, including footrope and headrope lengths, significantly influence the bycatch rate (Table 3 and Figure 10). Those characteristics influence the vertical opening of the trawl net. The loggerhead bycatch in the bottom trawl probably occurs during towing operations when the turtles are foraging on the bottom [16]. A higher vertical opening of the trawl nets during towing may increase the probability of a loggerhead catch. The longer the headrope, the larger the mouth of the trawl that is held open, increasing the probability of contacting sea turtles.

On-board observations and cameras set on the gear will be crucial to exploring the influence of trawl net openings on the sea turtle bycatch risk, and further testing of options related to mouth shape adjustment are needed to reduce the risk.

Previous studies have already highlighted the high impact of gillnets in Tunisia [24]. Gillnets could be considered passive fishing gear: turtles are caught by chance as they move from location to location. According to the fishermen, the turtles actively try to feed on fish entangled in the nets, damaging the gear. However, these nets may attract sea turtles trying to feed on targeted catches, thus increasing the probability of them being captured. The trammel nets and bony fish gillnets can be used with every kind of low-cost boat, and they allow big fish to be fished efficiently.

Even though the nets targeting bony fish, such as trammel nets and encircling nets, exhibit low BPUE (Figure 2), the number of fishing units using these types of gear is high in both gulfs; this represents a real threat and causes a significant level of bycatch. The quantification of total captures caused by these fishing units is very difficult to assess because of the high number of small boats involved along the Tunisian coastline which land outside the surveyed fishing ports. In fact, local and traditional use over the generations, together with the fishermen’s skill and the gear’s plasticity, has given rise to many variations which are very difficult to assess and classify [40]. Although a single net is unlikely to capture a turtle, the use of these types of gear on a large scale, especially near areas of high turtle density, can have a significant impact on the populations. Further studies are necessary. As previously reported, mesh sizes play a primary role [57,58,59]. The predicted values of the loggerhead turtles caught show that there is a significant difference between different types of gillnets (Figure 7). Ray nets and shark nets, the ones with the larger mesh sizes, show the highest BPUE and, therefore, represent a real threat to sea turtles. On average, the vessels operating with ray nets in Tunisia capture more than 15 turtles yearly, which is one of the highest reported capture rates in the Mediterranean.

Even with a low fishing effort, a vessel equipped with ray gillnets captures, within a year, as high a number of sea turtles as one trawler. The global impact of these gear types compared to the impact in Tunisia remains unclear since the national statistics about the number of vessels involved is not available. Shark and ray nets need particular attention and further studies for at least two reasons: they target sharks and rays as well as other vulnerable species, and they induce a high sea turtle bycatch rate, combined with a high mortality rate among the caught sea turtles (particularly with the ray nets).

For longlines, the hooks are technically defined by their shape, dimensions, material (steel, inox), and point (usually with a barb), as well as by the shape of the eye (flat or twisted). Based on the interviews, we only received information on hook size, and the GLMM detected a maximum risk for medium-size hooks, number #5 (SUN format), with a distance between the hook shank and the point of 1.8 cm. The bait type did not affect the predicted number of loggerhead turtles caught. In non-Mediterranean experiments (U.S. and Pacific data), mackerel bait reduced the turtle bycatch (82%) and increased the catch of swordfish compared to squid bait [60,61]. Ref. [20] demonstrated that stingray bait can reduce the catch rate of loggerhead turtles compared with mackerel bait. However, even if using stingray bait could reduce the bycatch rate of sea turtles [20], it cannot be considered a solution since stingrays are vulnerable species.

Levers of mitigation

This study provides a critical early step towards the reduction in the fisheries’ impact on sea turtles in Tunisia. The results suggest that the modification of the gear characteristics may reduce the sea turtle bycatch in Tunisia; this should include:-Testing modification of the trawl vertical opening and turtle excluder devices designed for fish trawls;-Testing deterrent devices, such as green LED lights on large mesh gillnets and trammel nets.

Legal measures, including the banning of shark and ray nets, should also be considered since they are doubly harmful in that they target vulnerable elasmobranchs and catch a high number of sea turtles.

Alternative fishery practices could be tested to replace or reduce the use of the trawls and nets that have proven to be the most impactful gear in Tunisia. Other gear types exhibiting a better selectivity could be proposed, such as traps or pots, hooks, and small purse seines.

In addition, this study highlights the bycatch seasonal patterns related to a particular gulf and the gear types that may orient the fisheries’ management towards reducing the risk of interaction with sea turtles, such as by spatiotemporal closure to trawlers in the Gulf of Gabes during summer or by the voluntary displacement of fishermen (move-on rule) when a high interaction risk has been identified.

Further collaborative works bringing together researchers, fishermen networks, and the national administration could lead to predictive management.

## 5. Conclusions

In the case of poor data, when resources are limited, involving and questioning fishermen and stakeholders is a good approach. The approach provided effective data with which to estimate the bycatch per unit effort and to identify the most impactful gear types, as well as the interaction hotspots and seasons [62]. In addition, it allowed us to detect the gear characteristics or practices influencing the bycatch risks and, therefore, to suggest future mitigation measures. It is also a good complementary approach to compensate for the risk of under-declaration associated with the mandatory reporting of the bycatch of vulnerable species required by law.

Interviews still have limits and need to be complemented by additional approaches, such as onboard surveys, particularly in assessing the bycatch mortality rate and the factors influencing it. The present results might still underestimate the real situation of the sea turtle bycatch in Tunisia. Ref. [39] confirms the fact that the typical fisherman reaction is to report low bycatch rates to avoid the restrictions that can be imposed by an administrative authority. However, fishermen should play a significant role in endangered species conservation efforts. The awareness of suitable methods for dealing with live caught turtles may also help to reduce the direct mortality. Awareness campaigns should be designed to inform fishermen about how to deal with captured turtles and how to apply recovery techniques to turtles.

The results presented in this study can be considered to be an important step towards further investigations on the sea turtle bycatch in Tunisian waters, and they identify mitigation measures that need to be tested in the future, including a change in the gear characteristics and the spatiotemporal management of the fishing effort.

In the short term, efforts aimed at increasing the survival of the captured individuals should be prioritized. This would include encouraging the reporting of incidental capture by fishermen and training them on how to safely release bycaught turtles. For example, it has been shown that simple reanimation measures implemented by fishermen on bycaught sea turtles before releasing them from trawls can drastically improve their post-release survival [63], thus significantly reducing the global impact of trawlers on these vulnerable species.

## Figures and Tables

**Figure 1 animals-13-01535-f001:**
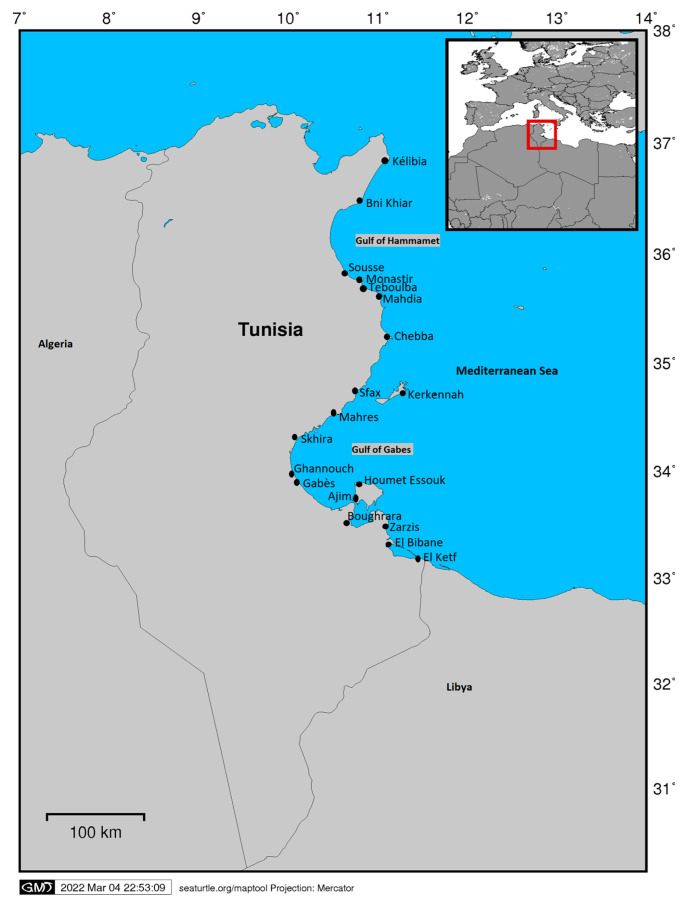
Map of the Tunisian ports where interviews were carried out (created with Maptool, SEATURTLE.ORG).

**Figure 2 animals-13-01535-f002:**
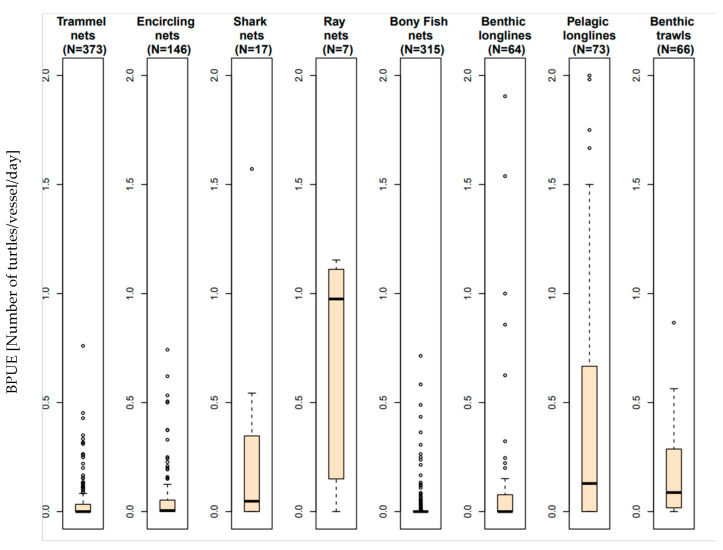
Sea turtle bycatch per day of fishing (BPUE).

**Figure 3 animals-13-01535-f003:**
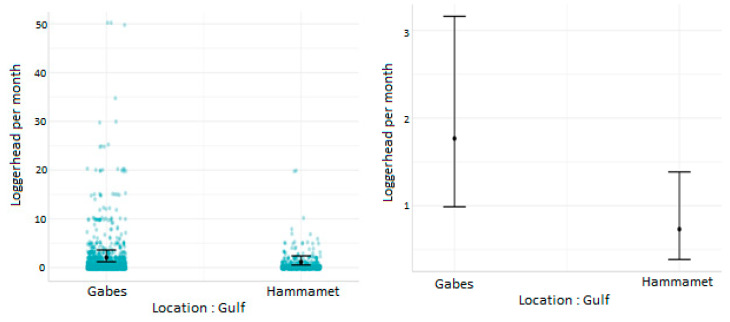
Number of sea turtles caught per vessel equipped with fishing nets per month in the gulfs of Gabes and Hammamet.

**Figure 4 animals-13-01535-f004:**
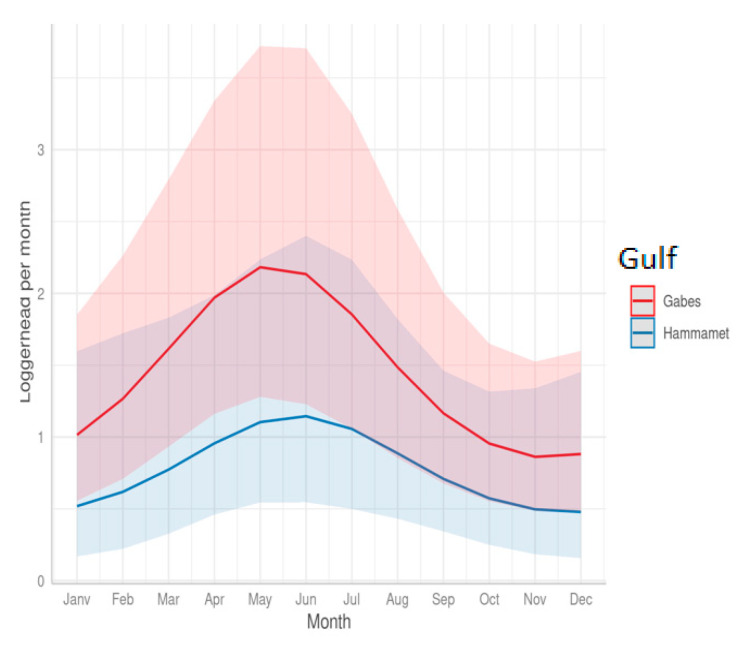
Fishing nets: monthly mean number of loggerhead turtles caught per vessel; predicted (ZINBmix) monthly number of loggerhead turtles in bycatch per fishing net by gulf.

**Figure 5 animals-13-01535-f005:**
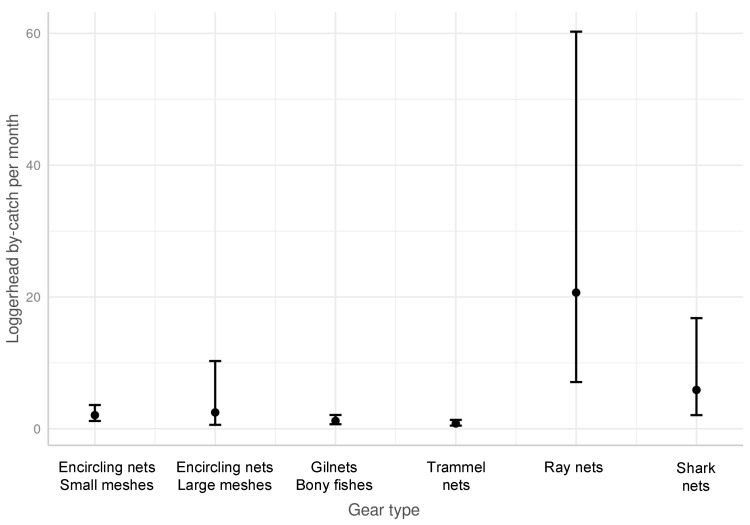
Marginal effect (ZINB mix) of gear type (nets) on predicted sea turtle bycatch quantity per month.

**Figure 6 animals-13-01535-f006:**
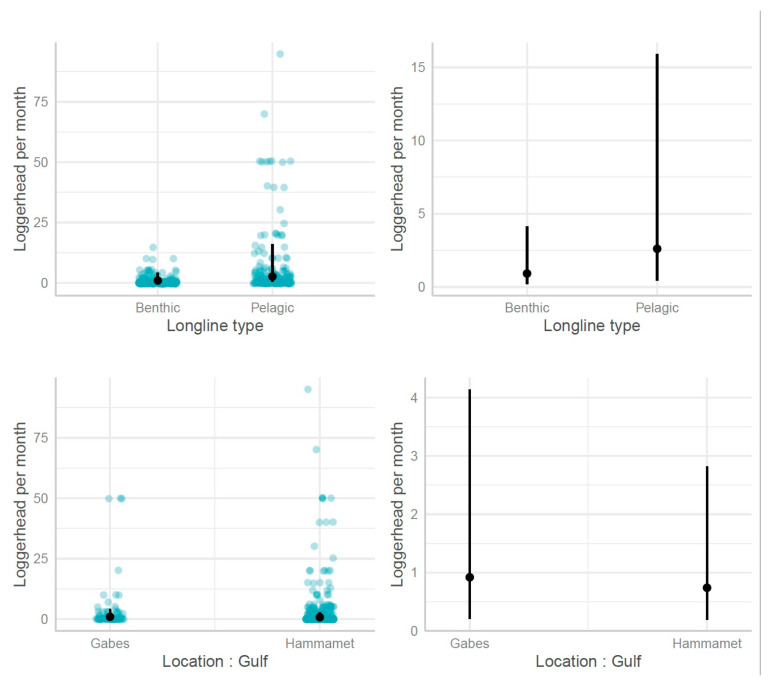
Predicted number of sea turtles per month per vessel equipped with longlines according to the longline type (benthic vs. pelagic) and location (Hammamet Gulf vs. Gabes Gulf): predicted and observed data; predicted mean and 95% CI.

**Figure 7 animals-13-01535-f007:**
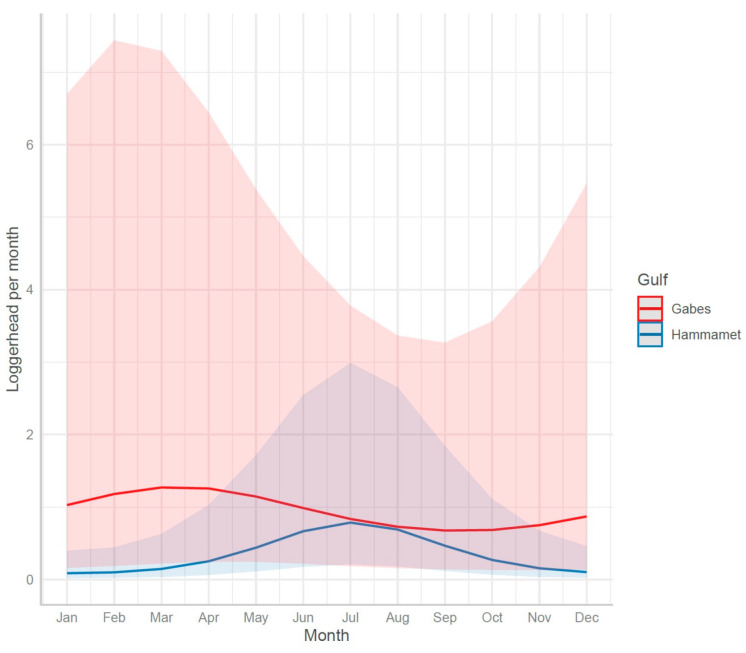
Seasonality (pseudo-Fourier terms in the GLMM model): annual variation of predicted monthly mean sea turtle bycatch quantity per longline vessel—Gabes vs. Hammamet.

**Figure 8 animals-13-01535-f008:**
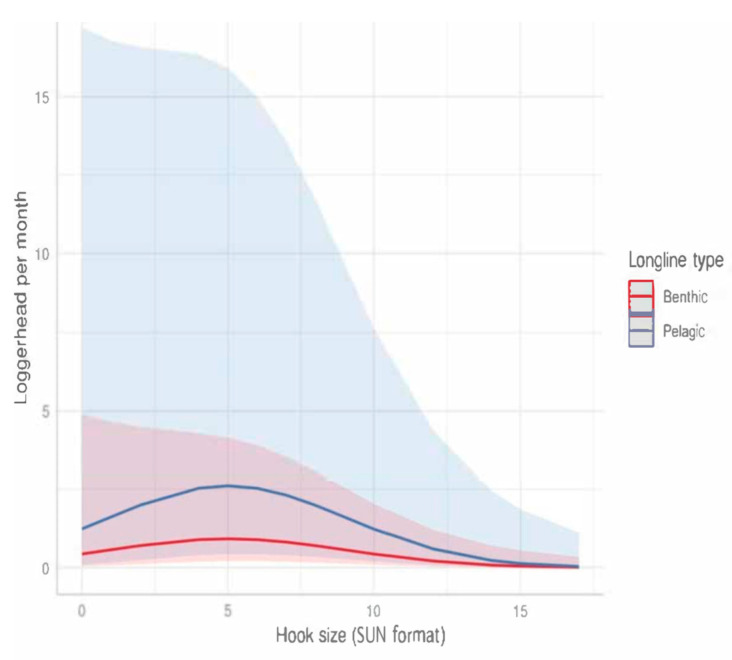
Marginal effect of hook size on predicted number of sea turtles caught monthly in longlines.

**Figure 9 animals-13-01535-f009:**
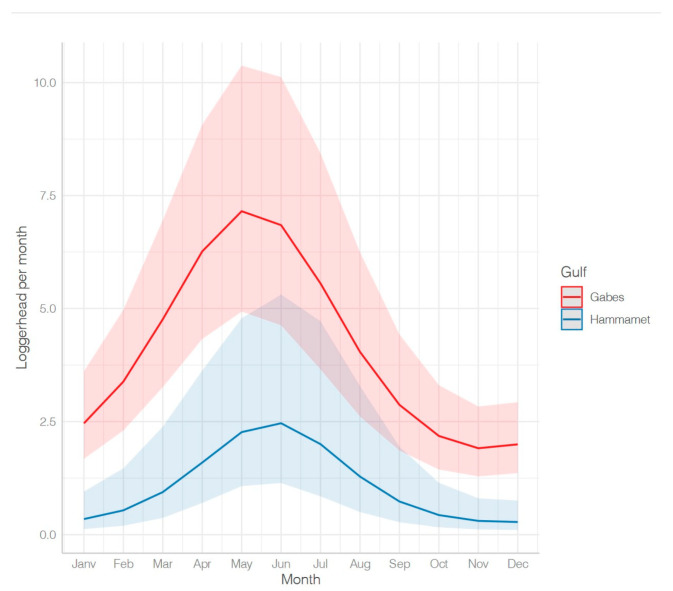
Predicted monthly mean sea turtle bycatch quantity per trawler—Gabes vs. Hammamet.

**Figure 10 animals-13-01535-f010:**
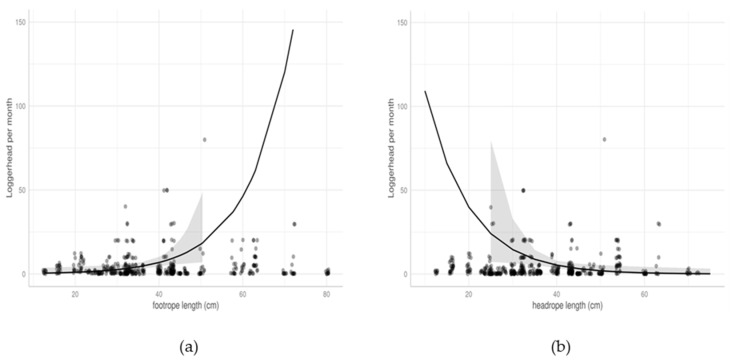
Effects of trawl characteristics on predicted monthly mean sea turtle bycatch quantity per trawler: (**a**) marginal effect of footrope length; (**b**) marginal effect of headrope length.

**Table 1 animals-13-01535-t001:** Extract: Zero-inflated negative binomial GLMM results—fishing net characteristics with significant effects on the bycatch rate.

	N_Loggerhead_Month		
Predictors	Incidence Rate Ratios	CI	*p*
Count Model			
(Intercept)	0.04	0.00–0.55	0.016
Type of nets [trammel nets]	0.39	0.22–0.70	0.002
Type of nets [Garrasia]	10.01	3.00–33.43	<0.001
Gulf [Gabes]*cos(PseudoFourier)	0.64	0.49–0.85	0.002
Observations	3795		

**Table 2 animals-13-01535-t002:** Zero-inflated negative binomial GLMM results—longline characteristics and seasonal factors with significant effects on the bycatch rate.

	Dependent Variable		
Predictors	Incidence Rate Ratios	CI	*p*
Count Model			
(Intercept)	0.00	0.00–0.00	<0.001
Longline type [pelagic]	2.84	1.44–5.59	0.003
Hook size [1st degree]	-	0.00–0.01	0.008
Hook size [2nd degree]	-	0.00–0.78	0.044
Gulf [Hammamet]*cos(PseudoFourier)	0.39	0.27–0.57	<0.001
Gulf [Hammamet]*sin(PseudoFourier)	0.56	0.40–0.78	0.001
Observations	429		

**Table 3 animals-13-01535-t003:** Extract: Zero-inflated negative binomial GLMM results—trawl characteristics with significant effects on the bycatch rate.

	Dependent Variable		
Predictors	Incidence Rate Ratios	CI	*p*
Count Model			
(Intercept)	0.01	0.00–0.09	<0.001
Gulf [Hammamet]	0.22	0.11–0.45	<0.001
Length of footrope	1.10	1.02–1.19	0.014
Length of headrope	0.90	0.83–0.98	0.017
Gulf [Hammamet] * cos(PseudoFourier)	0.34	0.18–0.62	<0.001
Gulf [Gabes] * sin(PseudoFourier)	1.29	1.05–1.58	0.014
Observations	637		

## Data Availability

Not applicable.

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
