# Peer review of "Fishermen Interviews: A Cost-Effective Tool for Evaluating the Impact of Fisheries on Vulnerable Sea Turtles in Tunisia and Identifying Levers of Mitigation"

_animals, 2023, doi:10.3390/ani13091535_

Round 1
Reviewer 1 Report (Previous Reviewer 1)
Thanks for the corrections and the manuscript ready better now.
Author Response
We appreciate you for your precious time in reviewing our paper and providing valuable comments. It was your valuable and insightful comments that led to possible improvements in the current version. The authors have carefully considered the comments and tried our best to address every one of them. We hope the manuscript after careful revisions meets your high standards. The authors welcome further constructive comments if any. Below we provide the point-by-point responses based on your comments and editors comments. We used the English editing service of the journal to improve the quality of our manuscript.

Reviewer 2 Report (New Reviewer)
The Ms is very interesting and helpful. Involving and questioning fishermen, this paper provides valuable data to estimate the bycatch of sea turtles in Tunisia and identify the most impacting impactful gear types and practices. Moreover, the obtained results allow for setting important mitigation measures and further study to improve knowledge on the impact of bycatch on endangered species.
It was tough to read and follow all the changes, but it's evident that the Ms was completely rearranged significantly improving the first version of the paper. Nevertheless, I found it too long. Some typos will be evident by clearing the last version and clearing all tracks (e.g. lines: 154, 157, 180).
Author Response
We appreciate you for your precious time in reviewing our paper and providing valuable comments. It was your valuable and insightful comments that led to possible improvements in the current version. The authors have carefully considered the comments and tried our best to address every one of them. We hope the manuscript after careful revisions meets your high standards. The authors welcome further constructive comments if any. Below we provide the point-by-point responses based on your comments and editors comments. We used the English editing service of the journal to improve the quality of our manuscript.

This manuscript is a resubmission of an earlier submission. The following is a list of the peer review reports and author responses from that submission.
Round 1
Reviewer 1 Report
Fishermen Interviews: a cost-effective tool to evaluate fisheries impact on vulnerable sea turtles in Tunisia and identify levers of mitigation by Louhichi et al.
They carried our 483 interviews were conducted in 19 ports and evaluated the turtle bycatch in Tunusia.
The manuscript is very good but needs some changes as given below.
1. The LC classification of loggerhead turtles in the mediterranean has been done in 2015.
2. The questionnare described well but can be given as supplement if they prefer.
3. Table 4 and Table 5 can be shortened or too big to follow (can be given only statistical different ones)
Discussion and conclusion can be shortened.
Author Response
We appreciate you for your precious time in reviewing our paper and providing valuable comments. It was your valuable and insightful comments that led to possible improvements in the current version. The authors have carefully considered the comments and tried our best to address every one of them. We hope the manuscript after careful revisions meets your high standards. The authors welcome further constructive comments if any. Below we provide the point-by-point responses. We used the English editing service of the journal to improve the quality of our manuscript.

Reviewer 2 Report
General
This is an interesting manuscript, presenting detailed information on sea turtle bycatch in Tunisia. However, at present, it is unwieldy, with far too much results – it is written like a thesis/report, not a paper – there should be 3-4 concise cohesive sections (1-2 paragraphs each) presenting the key findings of importance. Furthermore, throughout, tt needs very careful checking by a native English speaker throughout; I have identified issues in the Abstract and first paragraph of introduction; however, it is littered with errors – language, grammar etc, and would take too long to identify every single item; please ensure it is checked properly. Once all of these issues have been addressed, it could prove a useful contribution to the wider literature.
Simple summary
Title – this needs reconsideration, but only if you actually clarify in the abstract what these levers of mitigation are. There is nothing in the abstract on this, so, at present it must be deleted.
Line 14 – change “among the most impactful” to “represent the greatest threat”
Line 14 – the context of this sentence is unclear, “ It demonstrates interest of interview approaches”. Rather than “interest” do you mean “value?” Furthermore, interviews have been long used globally, so what is the novelty value of this in identifying mitigation approaches? This sentence needs careful restructuring.
Abstract
Line 17 “ In Tunisia, set nets, trawls and bait has an impact on sea turtles” – language issue here, bait what – or do you mean “baited gear” – if so clarify this? And it should be “have” not “has”
Line 18 – give context – X out of how many possible ports? Or if ports were selected based on region, clarify this.
Line 18 – this is not global, do you mean national or widespread, be careful of word use.
Line 19 – use simple consistent language, change metiers – use a layman term. Gear, as in the previous sentence?
Line 20, showed, be careful of tense use.
Line 21 – were, not are.
Line 21, typo, should be gear not gears.
Line 21 -in what way, what were the characteristics, locations and seasons of importance – this is too vague at present.
Line 22-24, any study could add this sentence; here you need to state how your specific findings could be applied, otherwise, you must remove this mitigation part from your title. What are the levers of mitigation – this is what the reader is waiting form
Introduction
This section needs careful revision, there are many small (even 1 sentence) paragraphs that do not connect or the context to your research focus is unclear. You need to group these into 3-4 larger paragraphs, with each containing a clear relevant message related to your focus. There are also multiple grammar/language issues throughout, I have identified some, but this needs very specific checking by a native speaker.
Line 29 – context of “really” incorrect. Replace with widely.
Line 30 – specially incorrect, especially.
Line 31 – artisanal should not be in capitals.
Line 32 – the reader has no idea what “it” refers to – clarify.
Line 34 -hosts, not is hosting
Line 35, give the scientific name for all three here.
Line 36, so do the other two species – this needs rewriting
Line 43-46, sentence too long and circuitous – needs revising.
Line 43-48 – the context of this paragraph to the focus of the paper is not clear. What is the relevance to your study? For instance, you could highlight the difficulty of obtaining accurate values on bycatch of immature and adult life stages hindering certainty in models used by IUCN Red data book. Hence the need to gather as much data as possible on bycatch.
Line 57-8 – this should not be a single paragraph, plus the sentence is not constructed properly. This should be with the IUCN component, as should the previous paragraph, but reorganised to build a clear argument.
Line 58, concern not concerned
Line 59-62, language/tense use issue.
Line 63 – this should not be a single paragraph.
Line 60, what requires to be improved – do not use They, clarify.
Line 70, I do not think you actually mean objectify – be careful of word use.
Line 73-82, this should be one paragraph, plus all sentences need careful revision.
Line 76 – state the factors – large diversity is vague.
Methods
Please ensure the language/grammar of this entire section is checked; I have only checked the content, not this, which should be done by an English language editor. There are sentence, language, grammar issues throughout – too many to list here.
There are also too many paragraphs, some 1 sentence long; these need to be grouped.
Line 85 – What do you mean by “All ports declared” – what is a declared port and by whom? What is considered a port – size, number of craft - more details to allow repetition is needed. How is a port distinguished from a harbour?
Also, state the number of ports, not just all.
Line 90, an interview is not led, implemented?
Line 91, simply state it was created in maptool, thanks is not needed.
Line 94, gear not gears.
Line 97, could not
Line 99, onboard their vessel
Line 105, fishing gear not gears
Line 111, clarify what you mean by last fishing trip, do you mean the trip immediately before interview? Here, you need to clarify the timeframe considered, i.e., within the last 24, 48 h or did you accept longer 1 week, 1 month? More?
Line 126 gear
Line 128 – what is a metier – this needs clarifying.
Line 136 variation
Results
Again, the grammar/language/sentence structure/tense use need careful revision throughout; I have not identified/looked at these items individually as there are too many to list. Also, paragraphs need restructuring.
This section is also excessively long – there should be 3-4 clear sections, with 3-4 supporting graphics. You need to reconsider what information you are presenting and why, and place the remainder as online supplement.
There are far too many figures and tables – there should be max 4-5 figures in total.
Line 164 - Why is total in capitals?
Line 164 – out of how many ports in Tunisia?
Line 167-8 – unnecessary repetition of methods
Line 178, why is R for ray in capitals here?
Line 179, not sure what you mean by come forth (maybe came fourth?). This needs rephrasing. Just say, followed by x, y, and z.
Figure 2 – do you actually mean number, or do you mean the average/mean? If you simply mean number, then you need to explain the SD/SE and mean lines, and you need to give total n for each, the actual number of turtles by actual number of vessels, somewhere on each graphic. Information is missing here to allow appropriate interpretation.
Table 1 – does this just repeat the graphic – only present a table or a graph, not both.
Line 191-196 – this paragraph is confusing, there is some information that belongs to the methods. It needs careful revising
Figure 3 – legend needs more detail.
Table 2 – again, if this repeats what is simply in the graph, delete.
Same applies for Figure 4 and Table 3
Line 209-216 – same applies
Line 222-224, this is methods, remove from here.
Line 227-228, you should not be citing other studies in the results
Line 236 – model design should be in methods section
Table 4 – should be placed as online supplement.
Figure 5 – you need to show these two gulfs on Figure 1 – to provide the reader with the necessary information.
Line 275 – bycatch, never bycatches
Line 278, use terms consistently, Gulf of Gabes
Line 275 and Figure 6a– which gulf for the bycatch (– were these both combined, if so why, when you have separated them to this point?
Line 285-88 – put in online supplement
Discussion
This is not set out appropriately for a manuscript. I have not checked it as the results section first needs extensive reshaping. Furthermore, it is not structured correctly.
There should be around 5-6 concise paragraphs
Para 1 – state your key findings, bringing them together cohesively – this sets the scene for each paragraph
Para 2 – consider key finding 1 with the wider literature
Para 3 – repeat for key finding 2
Para 4 – repeat for key finding 3
Para 5 – draw on the levers for mitigation -what they are and how they compare to the wider literature – this is a key element of your title, but does not actually materialise.
Para 6 - conclusion
Author Response

(The authors gave the same response as above.)

Round 2
Reviewer 2 Report
While the authors have made some changes, many remain superficial and uninformative, and have not actually addressed the comments raised; therefore, many comments remain unchanged. I have only checked the abstract to demonstrate this; please see the original comments throughout and check they have been addressed, simply changing phrasing is not addressing comments.
Line 14 – it is not clear how the value of interviews is confirmed; what is this assessed against as an alternative; this phrasing must change. The current way this is phrased is not true/appropriate.
Line 17 – this is a fact and requires citation; you need to be identifying what the knowledge gap you are addressing here is; as it stands, it implies the study has been answered before it has been done.
Line 18, as stated before, give context, how many main ports out of how many; it is only done on one part of the coastline actually, so not the whole of Tunisia, this is misleading. It is not national, as the north coast is missed.
Line 19-20, you need to clarify what was assessed here, or the context of the next sentence is not clear; what types of gear were assessed for these 3 to be the most common?
Line 21 – what are the other most harmful gear – the reader cannot follow these connotations.
Line 21-22, this remains too vague and uninformative; anyone could state this in an abstract. You need to state what characteristics, what locations and what seasons, as requested before; otherwise remove entirely from the manuscript.
Line 22-26, this sentence is too long and unclear, I could not follow it. Do not use pave to start with.
Introduction – the requested changes have not been made; this remains poorly formulated and not acceptable.
Methods – again the requested changes to paragraph structure have not been made
Results/Figures – the extensive requested changes have not been made
Discussion – as the previous sections have not been revised, this section remains inappropriate and the suggested changes not addressed.
Author Response
We appreciate you for your precious time in reviewing our paper and providing valuable comments. It was your valuable and insightful comments that led to possible improvements in the current version. The authors have carefully considered the comments and tried our best to address every one of them. We hope the manuscript after careful revisions meets your high standards. The authors welcome further constructive comments if any. Below we provide the point-by-point responses based on your comments and editors comments. We used the English editing service of the journal to improve the quality of our manuscript.
